# Disrupting the Molecular Pathway in Myotonic Dystrophy

**DOI:** 10.3390/ijms222413225

**Published:** 2021-12-08

**Authors:** Xiaomeng Xing, Anjani Kumari, Jake Brown, John David Brook

**Affiliations:** School of Life Sciences, University of Nottingham, Nottingham NG7 2UH, UK; xiaomeng.xing@nottingham.ac.uk (X.X.); mbxak13@exmail.nottingham.ac.uk (A.K.); stxjb15@exmail.nottingham.ac.uk (J.B.)

**Keywords:** myotonic dystrophy, CDK12, MBNL, RAN translation, molecular mechanism

## Abstract

Myotonic dystrophy is the most common muscular dystrophy in adults. It consists of two forms: type 1 (DM1) and type 2 (DM2). DM1 is associated with a trinucleotide repeat expansion mutation, which is transcribed but not translated into protein. The mutant RNA remains in the nucleus, which leads to a series of downstream abnormalities. DM1 is widely considered to be an RNA-based disorder. Thus, we consider three areas of the RNA pathway that may offer targeting opportunities to disrupt the production, stability, and degradation of the mutant RNA.

## 1. Myotonic Dystrophy: Phenotype and Mutational Basis 

Myotonic dystrophy is the most common and complex form of adult muscular dystrophy [1]. There are two different types, DM1 and DM2, both of which are caused by repeat expansion mutations. In this review, we will focus on DM1, which is characterized by myotonia, muscle weakness and wasting, and problems in a variety of other systems including the heart, brain, and endocrine [1]. In DM1, the (CTG)n repeat is located in the 3′ untranslated region (UTR) of the *DMPK* gene [2,3,4] whereas DM2 is caused by a tetranucleotide repeat expansion, (CCTG)n, located within an intron of the *CNBP* gene [5]. In each case, transcription of the mutant gene leads to expansion RNAs, which are retained in the nucleus, where they sequester muscleblind-like proteins, primarily MBNL1 and MBNL2, which regulate alternative splicing. Consequently, the splice-form distribution of several transcripts is affected. In effect, the RNA mutation pathway in DM can be broken down into three phases: transcription, transcript retention, and transcript liberation/degradation, each of which may provide targets for therapeutic intervention. In this review, we consider aspects of these three phases in the context of DM1 and possible therapies, along with the role of RAN translation in the final phase.

## 2. The Role of CDK12 in the Production of DMPK Expansion Transcripts

During transcription, the C-terminal domain of RNA polymerase II is subjected to multiple regulatory phosphorylation events that control its transcriptional potential. Several cyclin-dependent kinases are associated with the regulation of transcription including CDK7, CDK8, CDK9, CDK12, and CDK13 [6]. These kinases are important for transcriptional initiation and/or elongation. Cyclin-dependent kinase 12 (CDK12) is a transcriptional kinase comprised of 1490 amino acids, which has recently been shown to play a role in nuclear foci formation in DM1 [7]. It was first discovered in 2001 as a putative element of transcription process [8]. *CDK12* and *CDK13* are metazoan orthologs of the Saccharomyces cerevisiae gene *Ctk1* [8,9]. Both *CDK12* and *CDK13* have evolved by duplication and the proteins they encode share a 43% sequence similarity and a largely conserved kinase domain (KD) (>93%). Based on the interaction of CDK12 with overexpressed cyclin L, cyclin L was initially reported to be the regulatory subunit, and the same studies suggested a role in the regulation of alternative splicing. However, recent studies have reported that the endogenous human CDK12 does not associate with cyclin L but rather with cyclin K [10]. CDK12 and CDK13 bound to cyclin K operate in separate complexes in mammalian cells [8,10,11,12]. CDK12 preserves genome stability by promoting Ser2 phosphorylation at the carboxyl-terminal domain of RNA polymerase II (CTD), thereby facilitating transcription [8]. CDK13 plays a key role in modulation of the phosphorylation state and the activity of various splicing factors. The functional differences between CDK12 and CDK13 are possibly due to the different N- and C-terminal extensions [12].

*CDK12* transcripts are expressed in all human tissues [8] and CDK12 protein is essential in both adult and developmental phases [13]. CDK12 mRNAs are primarily expressed as two transcripts corresponding to a size of 8.5 and 6.8 kb [8].

## 3. CDK12 in RNA Processing

CDK12 was found to be located at the promoter sites of protein-coding genes. These sites usually overlap with RNA PII occupancy sites [14]. Hence, CDK12 helps in the elongation process by close association with RNAPII [8,12]. One of the most vital roles of CDKs is the phosphorylation of the carboxy-terminal domain (CTD) of RNA polymerase II (RNAPII). The CTD of RNAPII consists of multiple repeats of a conserved hepta-peptide consensus sequence Tyr-Ser-Pro-Thr-Ser-Pro-Ser. The number of these hepta-peptide repeats varies largely among different groups of organisms, ranging from 26 repeats in yeast to 52 in mammals [15,16]. The phosphorylation of CTD generates a complex regulatory code also known as the CTD code [17]. This regulates the transition of RNAPII from a hypo-phosphorylated form, which enables it to enter the preinitiation complex, to a hyper-phosphorylated form, which is capable of processive elongation of the transcript. Both CDK9 and CDK12 have been associated in the process of Ser2 phosphorylation in the heptad repeat leading to the release of paused RNAPII, thereby allowing productive transcript elongation [18].

Inhibitors of CDK12 have been developed and one such inhibitor, THZ531, when bound to CDK12 leads to the premature termination of transcript formation for genes involved in DNA repair and DNA replication [14], based on the length of the genes involved and the presence of many polyadenylation sites [19,20]. Cells from mouse models have shown that fully transcribed protein-coding mRNAs are downregulated by inducible inhibition of CDK12 [21]. CDK12 also plays a critical role in mRNA processing. Many splicing factors present in nuclear subdomains are substrates for CDK12 [20]. CDK12 uses two different mechanisms for mRNA processing: direct association with mRNA processing factors [8,22,23] and CTD phosphorylation-mediated recruitment of the factors involved in splicing and 3′ end processing [24,25]. CDK12 also modulates the alternative last exon (ALE) splicing process [26]. Examples of genes where RNA processing is regulated by CDK12 include SRSF1 [22] and c-MYC [27].

## 4. CDK12 as a Therapeutic Target for Myotonic Dystrophy

Medium-throughput phenotypic assays developed in our lab, based on the identification of nuclear foci in DM patient cell lines using in situ hybridization and high-content imaging, facilitated the identification of potentially useful therapeutic compounds [28]. Other studies have shown alleviation of the DM1 molecular phenotype by small-molecule ATP-binding site-specific kinase inhibitors [29]. The main target of one subset of these kinase inhibitors was identified as CDK12, which provides a novel druggable molecular target for myotonic dystrophy. Ketley et al. demonstrated the relationship between CDK12 and the production of mutant DMPK transcripts [7]. CDK12 inhibitors were found to eliminate the nuclear foci in DM cells as well as in the HSA^LR^ mouse model, leading to potentially beneficial effects [7]. 

Ketley et al. showed the level of CDK12 protein was elevated by 48% in DM1 muscle biopsies compared to healthy control muscle samples. Further immunohistochemistry experiments demonstrated co-localization of CDK12 with SC35 speckles in both DM1 and non-DM1 cells, consistent with previous studies [8]. Nuclear foci containing the mutant transcripts have also been reported to localize with the nuclear speckles [30,31]. Hence, the authors also quantified CDK12 granular structures, which again were seen to be significantly increased in DM1 cells when compared to non-DM1 cells. shRNA knockdown of CDK12 in DM1 cells produced a decrease in the amount of nuclear foci and CDK12 granules by 69% and 56%, respectively, whereas CDK12 overexpression in the DM1 cells resulted in increased numbers of nuclear foci [7]. The relationship between CDK12 and mutant transcripts was further validated by treating the cells with the selective CDK12 inhibitor THZ531 [14]. THZ531-mediated inhibition of CDK12 in DM1 cells affected the formation of mutant expansion transcripts more than wild-type transcripts. Thus, both inhibitor-mediated targeting and genetic knockdown of CDK12 led to a significant decrease in nuclear foci [7]. Using a bidirectional inducible plasmid expressing eGFP and 960 interrupted CTG repeats, in opposite directions, Ketley et al. showed that the primary effect of CDK12 inhibition was on transcription, rather than on transcript degradation [7] (Figure 1).

CDK12 is not involved at the initiation of the transcription process and its inhibition does not lead to global transcriptional arrest [19]. It is thought to be involved in the transcription of ‘long’ transcripts [9]. More work is needed to identify and test novel inhibitors that will help inform our understanding of CDK12 potency and selectivity. Given the significant relationship between CDK12 inhibition and elimination of nuclear foci, refined targeting of CDK12 may help us to develop a potential method to eliminate the RNA pathogenetic mechanisms in myotonic dystrophy.

The possibility of targeting transcription for therapeutic benefit in DM1 was raised by Siboni et al. [32], who showed that treatment with a low dose of the transcriptional inhibitor Actinomycin D rescued mis-splicing and reduced expansion RNA in a DM mouse model.

In addition to CDK12, other kinases have been suggested as potential drug targets in DM1. One such case is glycogen synthase kinase 3β (GSK3β) [33]. An increase in GSK3β activity was observed in the skeletal muscle of the DM model HSA^LR^ mice prior to the progression of skeletal muscle weakness [34,35]. The increase in GSK3β activity is caused by the repression of cyclin-dependent kinase 4 (Cdk4) signaling by the mutant expansion transcripts. A significant reduction in DM1 muscle pathology was seen when HSA^LR^ mice were treated with inhibitors of GSK3β [33]. Clinical trials are currently in progress with the GSK3β inhibitor AMO-02 (Tideglusib) [36].

## 5. MBNL, CELF, and Other AS Factors Involved in DM Pathology

One of the best-studied aspects of the molecular pathway underlying myotonic dystrophy is the nuclear retention of mutant expansion transcripts, resulting in the sequestration of MBNL proteins and activation of CELF1 to produce RNA mis-splicing in several different transcripts [37,38] (Figure 2).

CELF and MBNL proteins are antagonistic regulators of alternative splicing. In DM1, MBNL protein activity is repressed due to its sequestration by toxic RNAs, while CELF protein activity is upregulated by protein kinase C (PKC)-mediated CELF1 hyperphosphorylation [39] (Figure 2).

In addition to a role in alternative splicing regulation, it has been shown that MBNL plays a role in poly A site selection. Studies using HITS-CLIP and minigene reporter assays revealed that MBNL binding to the 3′ untranslated regions (3′UTRs) of target RNAs has a direct impact on polyadenylation site switching. MBNL-binding sites are frequently concentrated proximal poly(A) signals (pAs). Depletion of MBNL resulted in silencing of distal pAs while proximal pAs were boosted [40]. Thus, loss of MBNL misregulates alternative polyadenylation, affecting multiple pre-mRNA processing procedures in DM [40]. The upregulation of many alternatively spliced last exons in DM skeletal muscle-related transcripts was closely correlated with estimated MBNL function and consequently muscle strength [41]. MBNL protein sequestration and dysfunction causes a reversion to fetal polyadenylation patterns in adult muscle, which may induce protein synthesis inhibition and promote protein catabolism leading to muscle atrophy [38]. Likewise, CELF1 proteins bind to the 3′ UTRs of mRNAs and increase the density of upstream cleavage and polyadenylation sites by recruiting cytoplasmic deadenylases [42]. Downregulation of CELF1-bound mRNAs is dose dependent on CELF1 binding to 3′ UTRs [42]. CELF1 binding to alternative 3′ UTRs and alteration of 3′ UTR isoform abundance is associated with isoform-specific repression. The possible consequence is the altered regulation of cleavage and polyadenylation to produce isoforms with different pAs [42].

CELF1 and MBNL1 have similar binding sites in 3′ UTRs and tend to cluster together, where they compete to bind their co-target mRNAs and lead them to different fates. The extent of CELF1 and MBNL1 binding to 3′ UTRs correlates with the degree of mRNA suppression or stabilization, respectively. Greater MBNL binding prevents mRNAs from CELF-mediated inhibition. However, this antagonism effect by MBNL1 and CELF1 is also dependent on binding site proximity. Wang et al. (2015) showed that a distal MBNL1 binding site in 3′ UTRs was sufficient to prevent mRNA repression in the presence of a CELF1 site, but this protective response was not observed in the presence of a proximal MBNL1 site [42]. mRNAs with more proximal CELF1/MBNL1 pairs are more strongly downregulated than those with distal CELF/MBNL1 pairs [42]. However, no evidence has shown direct antagonism of the binding between CELF and MBNL proteins.

In addition, CELF and MBNL proteins antagonistically regulate alternative splicing of over 200 exons [42]. For instance, MBNL1 is responsible for the regulation of ABLIM1 (actin-binding LIM protein 1) exon11 and CLCN1 (muscle chloride channel) exon7a splicing, which are dysregulated due to sequestration of MBNL1 in the expansion RNAs in DM1. The CELF family also contributes to the aberrant ABLIM1 and CLCN1 splicing by antagonizing the effect of MBNL1 [43]. In both cases, aberrant splicing is associated with muscle symptoms of DM1. Many other alternative splicing events have been well validated to be MBNL dependent, including those affecting MAPT, NUMA1, MBNL2, NCOR2, and LDB3 [41]. The binding position of MBNL to its target RNAs determines the consequence of splicing events: exon inclusion or exon exclusion. MBNL binding within alternative exons and upstream intronic regions usually promotes exon skipping, while binding to downstream intronic regions promotes exon inclusion. This splicing regulation mechanism resembles that of other factors, such as neurooncological ventral antigen (NOVA), forkhead box (FOX), or heterogeneous nuclear ribonucleoprotein L (hnRNP L) [44].

It was discovered recently that another RNA-binding protein, heterogeneous nuclear ribonucleoprotein A1 (hnRNPA1), which is downregulated during normal postnatal development, remains elevated in DM1. Thus, it induces fetal splicing shifts of target RNAs in this disease. Reduced lifespan and muscle pathology were observed following AAV-mediated overexpression of hnRNPA1 in a mouse DM1 model [45].

TRIM71 is a member of the Trim-NHL protein family, which modulates alternative splicing by repressing MBNL1 in embryonic stem cells. It functions via binding to hairpin structures in the 3′ UTRs of MBNL1 mRNA [46]. Recent work has shown that TRIM71 represses the mRNA of CDKN1A/p21, a tumor suppressor and cell cycle regulator, in conjunction with nonsense-mediated decay factors SMG1, SMG7, and UPF1 [47]. It has also been shown that UPF1 associates with TRIM71 to regulate the expression of TRIM71 mRNA targets, which possess a 3′ UTR longer than 750 nt [47,48]. Interestingly, work with C. elegans points to a role for UPF1 in the degradation of mutant DMPK mRNA [49].

## 6. Domains and Sub-Cellular Localization Important for MBNL Function

MBNL proteins bind to toxic RNAs containing CUG and CCUG repeats through specific recognition of the Watson–Crick face of the GC dinucleotide [50], binding to the YGCY motifs (where Y indicates pyrimidines) with their four CCCH zinc fingers (ZnFs). The four ZnFs form two tandem ZnF pairs (ZnF1–2 and ZnF3–4), which differ in their ability to bind target RNAs and regulate splicing events. ZnF3–4 shows lower affinity binding to target RNA than ZnF1–2. Hale et al. (2018) designed two synthetic MBNL proteins with duplicate ZnF1–2 or ZnF3–4 domains, which brought the splicing activity up to 5-fold and down to 25%, respectively, compared to a natural MBNL protein [51]. Their findings suggest that ZnF1–2 drives alternative splicing activity through specific recognition of YGCY motifs, whilst ZnF3–4 serves as an aspecific RNA-binding domain, which assists MBNL to recognize a wide range of target mRNAs by binding to a broader region of motifs [51]. These results are consistent with previous studies showing truncated MBNL proteins containing only one of the two ZnF pairs bind to YGCY motifs impotently, suggesting that the tandem ZnFs work collaboratively [52].

However, other regions of the proteins may also interact with RNA and play an important role in regulating gene expression and RNA splicing [53]. Regulatory regions of MBNL for alternative splicing are located within an 80-amino-acid segment, downstream of the N-terminal zinc-finger pair [53]. These domains possess common features with the splicing regulatory regions within CELF proteins [53].

Studies by Kino et al. (2015) and Lee et al. (2013) suggested cross-regulation and splicing compensation among MBNL proteins [54,55]. Overexpression of any MBNL paralogs repressed the inclusion of exon 5 (54nt), which is essential for the nuclear localization and splicing activity of MBNL. A reduction in one of the MBNL paralogs would be compensated for by the increased nuclear isoforms of the other MBNL protein. Lee et al. (2013) also found that MBNL2 can functionally compensate for the loss of MBNL1 by increased binding to MBNL1-related splicing targets, resulting in partial preservation of the adult splicing pattern [55]. 

Although nuclear localization of MBNL is crucial for its alternative splicing activity, Pei-Wang et al. (2018) showed that it is the MBNL1 cytoplasmic isoform, not the nuclear isoform, that promotes neurite morphogenesis and reverses the morphological defects caused by CUG triplet repeat RNA [56]. Expanded CUG RNA induces de-ubiquitination of cytoplasmic MBNL1, which leads to nuclear translocation and defects in neurite outgrowth and differentiation. Inhibition of MBNL1 de-ubiquitination with o-PA (1,10-phenanthroline), an inhibitor protecting K63-linked polyubiquitin chains from decay, reduces RNA foci formation and remarkably preserves the morphological features of neurons expressing CUG repeat mRNA [56]. Moreover, MBNL1 exon7 (36nt) is fundamental for MBNL dimerization [57,58]. MBNL1 +ex7 proteins are required for the abundance and splicing of specific mRNAs involved in cell migration and division and DNA repair [57].

## 7. Therapies to Alleviate MBNL Sequestration or Overexpress MBNL

Oligomeric compounds and other small molecules have been applied to specifically target the DM1 pathobiology at the DNA or RNA level via selective recognition of the repeating array of 1 × 1 nucleotide U/U internal loops formed by (CUG)^exp^ RNA. It has been shown that a dimeric 2,9-diamino-1,10-phenanthroline derivative (DDAP) designed by Li et al. (2018) [59] and monomeric bioactive ligands designed by Angelbello et al. (2021) [60] enable interaction with (CUG)^exp^ RNA and inhibit the formation of (CUG)^exp^ RNA-MBNL1 complexes. Those molecules are dimers that contain two 5′CUG/3′GUC-binding modules. DDAP binding to (CUG)^exp^ RNA is enhanced through Π–Π stacking interaction with neighboring guanines [59]. Amongst the ligands, the doubly guanidinylated molecules are the most effective inhibitors of the (CUG)^exp^ RNA-MBNL1 complex by efficient interaction with the phosphodiester backbone [60]. Some of these compounds significantly ameliorated DM1-associated molecular defects including toxic CUG repeat RNA aggregation into foci and alternative splicing deficiency [59,60]. Similarly, Lee et al. (2019) designed multivalent ligands, which effectively blocked transcription of d(CTG·CAG)^exp^ and inhibited the (CUG)^exp^ RNA-MBNL1 interaction [61,62].

Chakraborty et al. (2018) identified a drug, daunorubicin hydrochloride, which was reported to interact with double-stranded RNA [63]. Their research indicates that daunorubicin directly binds to CUG RNA repeats and alleviates MBNL sequestration. Similarly, erythromycin, which is a widely used antibiotic to treat various bacterial infections, has been shown to dissipate RNA foci and rescue missplicing in DM1 cell models by displacing sequestered MBNL [64]. Another potential approach to alleviate MBNL sequestration is by using oligopeptides. Cpy(a/t)(q/w)e D-Hexapeptides compete with MBNL binding to CUG repeat RNAs as well as doubling the transcription level of MBNL1. As a consequence, MBNL-dependent missplicing events are rescued. Intriguingly, those peptides share some similarities in structure and sequence with conserved amino acids within ZnFs of MBNL [58].

Aside from MBNL release, regulating MBNL expression intrinsically has also be investigated to ameliorate DM1 symptoms. Overexpressing MBNL1 improves the proliferation capability of skeletal muscle satellite cells (SSCs) in DM1 by inhibiting autophagy via the mTOR pathway [65]. Treatment with the anti-autophagic drug chloroquine was shown to upregulate MBNL1 and 2 proteins in animal DM models and patient-derived myoblasts, alleviating myotonic dystrophy phenotypes [66]. It was also found that MBNL2 expression was significantly upregulated in breast and lung cancer cells following treatment with a natural compound, Neobractatin (NBT), and overexpression of MBNL2 inhibited tumor metastasis [67].

Researchers recently discovered that tumor necrosis factor-alpha (TNF-α), which is a proinflammatory cytokine and a crucial contributor to various metabolic syndromes, modulates the expression of MBNL [68,69]. MBNL1 and 3 exhibited a significant reduction at the transcriptional level in cells under TNF-a-induced metabolic stress [69]. Previous studies showed high glucose induces TNF-α expression [70] and MBNL1 expression was reduced under high-glucose conditions [71], which strengthens the role of TNF-α in regulating MBNL expression. Concordantly, the activation of cyclooxygenase (COX) stimulates intracellular signals via the NFκB, p38, or MAPKs signaling pathway, which modulates proinflammatory cytokine expression, such as TNF-α, interleukin 1 beta (IL1β), and interleukin 6 (IL6) [72]. It was found that Phenylbutazone and other COX-1-selective nonsteroidal anti-inflammatory drugs (NSAIDs) could upregulate MBNL1 expression in myogenic cells via COX-1 inhibition [73,74]. COX-1 inhibition elevated MBNL1 transcription by suppressing methylation of a specific region in MBNL1 intron 1. 

## 8. Other Potential Therapeutic Interventions 

Previous investigations have explored the potential of targeting the C(C)UG repeat sequence directly using antisense oligonucleotides (ASOs). For example, phosphoroamidate morpholino oligomers (PMOs) comprise a short segment of CA(G)G repeats that bind to expanded C(C)UG^exp^ RNA directly [75]. However, those PMOs also target DMPK transcripts from the wild-type *DMPK* allele or other alleles containing C(C)TG repeat expansion, which may induce loss-of-function phenotypes, such as myopathy [76].

An alternative approach to reducing the expression of expansion RNA involves the application of microtubule inhibitors. Kaalak Reddy et al. (2019) identified multiple microtubule inhibitors, which selectively reduced CUG expansion RNA levels and partially rescued DM1-associated missplicing [77]. Treatment with one of the microtubule inhibitors, colchicine, selectively inhibited the generation of CUG expansion transcripts rather than enhancing their rate of turnover or degradation [77]. The microtubule-dependent transport of molecules into the nucleus can regulate chromatin organization and transcriptional profiles via LINC (linker of nucleoskeleton and cytoskeleton) complex bridges [78]. Combined knockdown of SUN1/2, the core components of LINC complex, produced a modest but significant reduction in the relative CUG repeat RNA levels [77]. Their findings indicate a potential role of microtubule dynamics in CUG expansion RNA transcription, which may be at any rate partially driven by interactions with the LINC complex.

In addition to the small molecules mentioned above, a variety of other small molecules have been developed or are being developed for use in DM therapy. It is beyond the scope of this review to mention them all, but two reviews provide a great deal of detail about such small molecules [79,80].

## 9. RAN Translation

In Section 10 of this review, we will consider the fate of the mutant expansion transcripts following their nuclear retention. In this regard, Repeat-Associated Non-AUG (RAN) translation is a well-documented and seemingly common method of action for diseases involving expanded repeat sequences, such as those seen in both myotonic dystrophy type 1 and 2. For a long time, it was believed that many of these expanded sequences would have no impact on translation and protein production as they were located in intronic or untranslated regions. In 2011, Zu et al. discovered that neither inhibition of the promoter sequence nor insertion of a stop codon cassette, upstream of a CAG_n_-containing minigene, were able to prevent the translation of three polypeptides formed from the three reading frames of the repeat [81]. Subsequently, it was demonstrated that RAN translation was highly dependent on the repeat length, with CAG sequences of less than 42 noted to not produce RAN-derived polypeptides [81]. RAN translation has been observed to occur bidirectionally in several repeat expansion diseases including DM1 [82,83,84]. Consequently, bidirectional transcription in combination with multiple reading frames presents the possibility of trinucleotide repeat diseases producing up to six potentially toxic polypeptides simultaneously.

Despite it now being well categorized that RAN translation exists and contributes to the pathology of several repeat expansion diseases, the underlying mechanism that drives this form of translation remains largely a mystery, although insights have developed via similarities and differences with the canonical translation pathway. The main stages of eukaryotic translation are as follows: eukaryotic initiator factor 2 (eIF2) forms a ternary complex with met-tRNA. This then forms the 43s preinitiation complex with the 40s ribosomal subunit and several other eIFs. In parallel, mRNA is identified by more eIF proteins by way of its highly methylated 5′ cap and the downstream region is unwound to promote translation. The entire complex then scans the mRNA 5′-3′ for the AUG start codon in order for translation to commence [85]. 

RAN translation of the repeat sequence CGG has been shown to be dependent on both the 5′ m^7^G cap as well as ribosomal 5′-3′ scanning to find an initiation codon, and that the initiation codon used is inconsistent between reading frames [3]. These data support the idea that RAN translation and canonical eukaryotic translation share, largely, the same mechanism. Both rely on mRNA cap recognition and the action of many eIF proteins to support unwinding of mRNA and scanning to find the desired start codon. Consequently, it is likely that the main difference between RAN and canonical translation lies within the method of initiation codon selection, although the exact way that this works is unknown [3]. As RAN translation has also been demonstrated to be highly dependent on repeat length, it is possible that the hairpin structures formed by long repeat expansions [5] play a pivotal role in initiation.

Although similar in many ways, the repeat expansion involved in DM1 and DM2 presents a key difference in RAN translation. DM1, being a trinucleotide CTG expansion [5], will theoretically produce six different poly(n) proteins due to the nature of the triplet reading frames of translation. Alternatively, for DM2, being a tetranucleotide CCTG expansion [5], it does not produce homogenous peptides, rather the proteins produced will be poly(4n) (Figure 3).

Of the five different polypeptides shown for DM1 in Figure 3a, only poly(gln) from the antisense strand has been identified in animal models and human disease tissue. Zu et al. demonstrated, using a mouse model of DM, reproducible poly(gln) nuclear aggregates in cardiac myocytes as well as a more frequent presence in leukocytes [81]. In DM1 patient samples, poly(gln) was identified at varying levels in myoblasts, skeletal muscle, and blood [81]. These poly(gln) aggregates are known to co-localize with caspase-8 [81] and caspase-10, presenting a possible method of toxicity through activation of a caspase-mediated apoptotic pathway [86].

Unlike DM1, both the sense and antisense RAN polypeptides have been identified in DM2. Zu et al. used immunostaining techniques to demonstrate the presence of α-LPAC (sense) and α-QAGR (antisense) polypeptides in brain tissue samples from DM2 autopsies, although with significant differences between the two proteins. Transfected cells and autopsy tissue both showed distinct localization of the proteins: α-LPAC localizing to the cytoplasmic regions and α-QAGR localizing to the nuclear regions. Both proteins formed small aggregates in their respective regions [87]. Furthermore, as well as different cellular localization, both proteins were observed to be translated in distinct types of brain tissue. α-LPAC protein was seen in the gray matter regions of the brain, notably pyramidal cells of the cornu ammonis and granular neurons of the dentate gyrus as well as neurons of the cortex and striatum [87]. In contrast, α-QAGR proteins were seen primarily in the white matter regions of the brain. Both RAN proteins were also seen in macrophages, although the study did not conclude whether these were a result of RAN translation within the macrophages or were ingested by phagocytosis. 

Finally, Mbnl1 has been implicated as a direct modulator of some RAN translation in DM2. Zu et al. found that Mbnl1 co-transfection with CCUG reduces the amount of α-LPAC formed, and similarly that Mbnl1^−/−^ Mbnl2^−/−^ mice showed significant upregulation in RAN translation [87]. These findings were absent for the antisense strands, suggesting that sequestration of Mbnl1 by expansion mRNA provides a direct block to RAN translation, likely by physically preventing access to the mRNA.

## 10. Conclusions

In this review, we considered three phases of the repeat expansion mutation pathway that provide a focus for therapeutic intervention. The three phases: transcription, transcript retention, and degradation, are all currently under investigation as part of drug development efforts. Thus far, encouraging results have been produced in cell and animal models, but their application to patients and successful clinical outcomes are eagerly awaited.

## Figures and Tables

**Figure 1 ijms-22-13225-f001:**
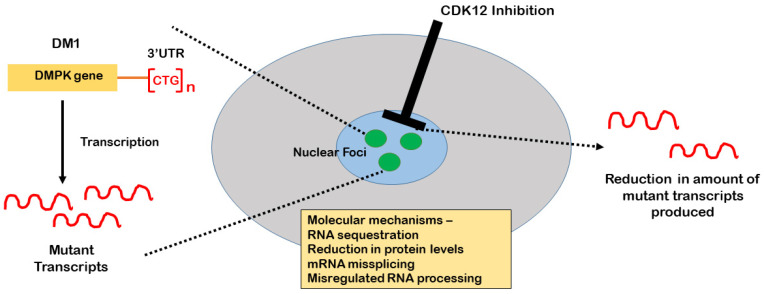
Diagram showing the mechanism by which CDK12 inhibition leads to a reduction in nuclear foci: Inhibitor-mediated targeting of CDK12 primarily targets transcription, leading to reduced mutant transcripts.

**Figure 2 ijms-22-13225-f002:**
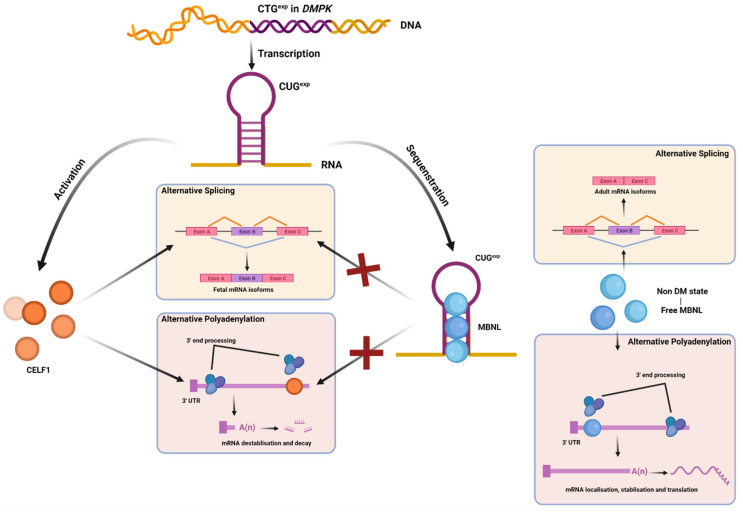
The misregulated alternative splicing and misregulated alternative polyadenylation model in human DM1. CUG^exp^ mRNAs sequester muscleblind-like (MBNL) and activate the expression of CUGBP Elav-Like Family Member 1 (CELF1) to a steady-state level through protein kinase C (PKC)-mediated hyperphosphorylation. These changes cause spliceopathy and fetal splicing shifts in adult tissues. In addition, CELF1 binding to 3′ UTRs leads to mRNA repression and decay while MBNL binding to 3′ UTRs promotes mRNA stabilization, localization to membrane compartments, and translation.

**Figure 3 ijms-22-13225-f003:**
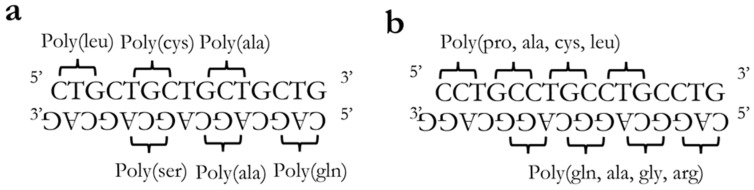
Theoretical polypeptides produced in (**a**) DM1 and (**b**) DM2. Currently, not all polypeptides shown have been identified in disease samples.

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
