# Peer review of "Disrupting the Molecular Pathway in Myotonic Dystrophy"

_ijms, 2021, doi:10.3390/ijms222413225_

Round 1

Reviewer 1 Report

The current article by Xing et al provides a careful and concise review of drug targets and potential therapeutics against DM1, a neurodegenerative disease caused by abnormal expansion of CTG trinucleotide repeats. The authors have discussed possible molecular mechanisms and biological consequences of disease development. This article focuses primarily on targeting toxic RNA, CDK12, and MBNL proteins involved in the pathogenesis of the disease. The authors carefully reviewed several important studies on these targets by small molecule compounds. Overall, this review summarizes several potential ligands capable of affecting DM1. Although the authors mentioned several important cases related to potential small molecule ligands targeting DM1, there are many studies in the literature that were not covered in this article. For example, JuYeon Lee et al. recently identified multi-targeting ligands for the treatment of DM1 (PNAS, 2019) or compounds targeting DM1-related toxic RNA (Lien Nguyen et al. 2015 JACS). Li J. et al. also reported a potential DM1 treatment compound capable of interfering with MBNL and RNA binding (Li L et al. Chemistry, 2018).

To make the current article more comprehensive for a wide readership, I would recommend the authors to add more studies on different small molecules that show potential to treat DM1. Although the figures in this manuscript are well elaborated and explanatory, it would be beneficial to summarize (perhaps in a table) a list of the compounds discussed with their potential target and proposed mechanism of action.

Author Response

Review Report 1

The current article by Xing et al provides a careful and concise review of drug targets and potential therapeutics against DM1, a neurodegenerative disease caused by abnormal expansion of CTG trinucleotide repeats. The authors have discussed possible molecular mechanisms and biological consequences of disease development. This article focuses primarily on targeting toxic RNA, CDK12, and MBNL proteins involved in the pathogenesis of the disease. The authors carefully reviewed several important studies on these targets by small molecule compounds. Overall, this review summarizes several potential ligands capable of affecting DM1. Although the authors mentioned several important cases related to potential small molecule ligands targeting DM1, there are many studies in the literature that were not covered in this article. For example, JuYeon Lee et al. recently identified multi-targeting ligands for the treatment of DM1 (PNAS, 2019) or compounds targeting DM1-related toxic RNA (Lien Nguyen et al. 2015 JACS). Li J. et al. also reported a potential DM1 treatment compound capable of interfering with MBNL and RNA binding (Li L et al. Chemistry, 2018).

We thank reviewer 1 for these comments. We have modified the manuscript to include the studies mentioned above.

To make the current article more comprehensive for a wide readership, I would recommend the authors to add more studies on different small molecules that show potential to treat DM1. Although the figures in this manuscript are well elaborated and explanatory, it would be beneficial to summarize (perhaps in a table) a list of the compounds discussed with their potential target and proposed mechanism of action.

It was not our intention to provide a comprehensive review of small molecule treatments for DM1 as this has been generated recently by ourselves and others [1, 2]. However, we understand the need to provide information that can be used by others, so we include references to recent papers describing many potential small molecule treatments not mentioned further in this review.

References

  1. Lopez-Morato, M., J.D. Brook, and M. Wojciechowska, Small Molecules Which Improve Pathogenesis of Myotonic Dystrophy Type 1. Front Neurol, 2018. 9: p. 349.
  2. Nakamori, M., Therapeutic approach for myotonic dystrophy: Recent advances in translational research. Neurology and Clinical Neuroscience.

Reviewer 2 Report

This review describes the main molecular pathways of Myotonic Dystrophy type 1 (DM1) pathogenesis. The authors also describe potential therapeutics that could be used to correct DM1 pathology including correction of CDK12 kinase. While this manuscript would be interesting for the broad audience, there are several minor questions that need to be addressed. 

  1. The first chapter of the manuscript is entitled “The role of CDK12 in the production of DMPK expanded transcripts”. However, this chapter contains only one sentence about CDK12 in DM1. Since the authors begin the paper with the role of CDK12 in DM1, it would be beneficial for the readers to include an introduction about DM1, the role of the mutant transcripts in DM1 and describe why kinases, including CDK12, are important in DM1 pathology. Such Introduction would provide a solid background for the chapter focused on CDK12.
  2. A significant portion of the manuscript is connected to CDK12 kinase in DM1 pathogenesis and its role as new therapeutic target. There are reports that showed that GSK3b-cdk4 pathway plays a role in DM1 and the inhibitor of GSK3 is currently used in the active Phase III clinical trial for pediatric DM1.  Therefore, it would be important to include some information about the role of GSK3b kinase as therapeutic target in DM1 pathogenesis.
  3. A brief conclusion at the end of the manuscript would be important to summarize the readiness to treat DM1 pathways.

Round 2

Reviewer 1 Report

The authors have made substantial corrections based on two reviewers' comments. I think the manuscript is suitable to publish.